# Prevalence and factors associated with multidrug resistant *Escherichia coli* carriage on chicken farms in west Nile region in Uganda: A cross-sectional survey

Ceaser Adibaku Nyolimati[1]*, Jonathan Mayito[1], Emmanuel Obuya[2], Atim Stella Acaye[3,4], Emmanuel Isingoma[3], Daniel Kibombo[1], D. M. Byonanebye[1,5], Richard Walwema[1], David Musoke[6], Christopher Garimoi Orach[5], Francis Kakooza[1]

1 Department of Global Health Security, Infectious Diseases Institute, Makerere University, Kampala, Uganda, 2 Department of Epidemiology and Biostatistics, Makerere University School of Public Health, Kampala, Uganda, 3 Department of Animal Health, National Animal Disease Diagnostics and Epidemiology Center, Ministry of Agriculture, Animal Industry and Fisheries, Kampala, Uganda, 4 Medical Research Council/Uganda Virus Research Institute & London School of Hygiene and Tropical Medicine (MRC/UVRI & LSHTM), Entebbe, Uganda, 5 Department of Community Health and Behavioural Sciences, Makerere University School of Public Health, Kampala, Uganda, 6 Department of Disease Control and Environmental Health, Makerere University School of Public Health, Kampala, Uganda

* cnyolimat@gmail.com

## Abstract

Infections with multi-drug-resistant (MDR) pathogens in food-animals threaten public health and food security. However, the epidemiology and factors associated with MDR *Escherichia coli* (MDR *E. coli*) on Ugandan farms are not well known. This study investigated the prevalence, resistance patterns and factors associated with MDR *E. coli* carriage on chicken farms. *Escherichia coli* was isolated from each of the 158 chicken farms sampled. The disc diffusion method for susceptibility testing was performed. Clinical breakpoints were interpreted according to Clinical and Laboratory Standards Institute guidelines. MDR was defined as resistance to three or more classes of antibiotics. MDR *E. coli* prevalence on chicken farms was 62.7% (95% CI: 55.0–70.3). High resistance was observed against ampicillin 79.8% (95% CI: 72.7–85.4), tetracycline 72.8% (95% CI: 65.2–79.2), cotrimoxazole 55.7% (95% CI: 47.8–63.3), and ciprofloxacin 38% (95% CI: 30.7–45.9). Male farm managers (Adjusted prevalence ratio [APR] = 0.72, CI: 0.55–0.93), attainence of at least secondary education (APR = 0.64, CI: 0.46–0.88) and administration of recommended antibiotic doses (APR = 0.76, CI: 0.59–0.96) posed a lower risk of MDR *E. coli* carriage while farms without footbaths posed a heighten risk (APR = 1.48, CI: 1.16–1.88). MDR *E. coli* carriage was highly prevalent on chicken farms in Uganda. This study underscores the urgent need for antimicrobial stewardship and improved infection prevention strategies on chicken farms.

**Data Availability Statement:** All relevant data are within the article and its Supporting Information files.

**Funding:** CAN, RW, JM and FK were supported by the Fleming Fund Country grants programme. This programme is funded by the UK Department of Health and Social Care, through Mott MacDonald to Infectious Diseases Institute, College of Health Sciences, Makerere University, Kampala under the terms of Grant Agreement|Uganda CG2 TOR FF78-484. The statements made and views expressed are solely the responsibility of the authors. The funders had no role in study design, data collection and analysis, decision to publish, or preparation of the manuscript.

**Competing interests:** The authors have declared that no competing interests exist.

## Introduction

Antimicrobial resistance (AMR) in foodborne pathogens poses a significant hazard to public health worldwide [1]. The demand for food-animals is high especially poultry, and is projected to increase by 17.8% by 2030 [2]. Globally, chickens account for 91% of the world's total poultry population [3]. In Uganda, over 85% of the livestock farmers rear poultry for livelihood and as a means of income generation [4]. Poultry is one of the main avenues for AMR emergence and transmittion of AMR genes in the food chain to humans [5]. AMR directly affects global economic growth due to costs associated with infections by resistant pathogens, with developing countries in Africa bearing the biggest burden of its negative effects [6, 7].

Unlike other antimicrobials, a continuous emergence and rapid dissemination of resistance against antibiotics among pathogens has been reported [8]. The increase is mainly attributed to the irrational use of antibiotics in animal [9]. Several risk factors contribute to the emergence of AMR including inadequate veterinary healthcare, lack of monitoring and regulatory services, excessive intervention by informal animal health service providers, and farmers' knowledge gap on AMR, which have resulted in the misuse and overuse of antibiotics in all types of animal farming [10]. Non-therapeutic use of antibiotics for growth promotion, and prevention and control of disease on poultry farms further predisposes to AMR emergence in animals [11, 12].

Resistant microbes may be disseminated directly or indirectly between animals and humans when exposed [13]. Cultural norms such as gender roles and management systems in animal production and close interactions between humans and animals have been reported as key drivers for AMR spillover and amplification especially in poultry [14]. Uganda developed a 5-year AMR National Action Plan (NAP in 2018), which sets out a framework of actions to address AMR within the country using a One Health approach [15]. However, laxity in policy implementation, unskilled human resources, and weak surveillance systems for AMR are underlying gaps in AMR control in Uganda [14]. Other factors associated with human exposure to resistant pathogens include proximity to poultry, shared water source, poultry wastes as fertilizer and gender roles [16]. Gender determines the division of labour in many contexts especially in Low- and Middle-Income Countries (LMICs) including animal rearing. In most farming communities in Sub-Saharan Africa (SSA), women, in particular, are responsible for caring for sick animals that are not intended for slaughter, while men majorly own the farms [17]. Compared to the male farmers, females farmers also plays critical roles in household activities and taking care of children including household water, sanitation and hygiene (WASH) activities [18]. These roles increases their exposure to and spread of resistant zoonotic pathogens including the drug resistant ones within female managed farms [17]. Increase in drug resistant animal-derived pathogens that threaten animals and humans health [19], can result from exchange of resistance genes between commensal and pathogenic microbes [20].

*Escherichia coli* (*E. coli*) is one of the commensal microbes within the gastrointestinal tract of animals [21]. However, strains known to cause illnesses in animals and humans have emerged in the recent past years [22], likewise resistant ones. *E. coli* play an important ecological role and can be used as a bioindicator of AMR [23]. Transmission of foodborne resistant pathogens from reservoirs, particularly poultry, to the human population does occur [24]. A study in Tanzania showed that resistant *E. coli* are widely distributed among humans, animals, and environment [25]. In Uganda, a high prevalence of MDR *E. coli* (88.4%) in clinically sick poultry has been reported [26]. Monitoring and surveillance of resistant pathogens across the human, animal and environmental interface is one of the best approaches for decision-making and reducing AMR [27]. In 2017, the World Health Organization (WHO) published a list of 12 priority antibiotic-resistant pathogens, among which was *E. coli* [28].

Poultry production accounts for one-third of global meat production [29] and poultry are among food-animals often raised under intensive conditions utilizing large amounts of antimicrobials, a practice that is driving AMR [30]. Despite the rapid growth of poultry production, extensive use of antibiotics and the rapidly increasing population, limited data exists on the prevalence, resistance patterns and the risk factors associated with MDR *E. coli* in Uganda. These limited data were predominantly sourced from the central [31, 32] and western [33] regions of Uganda with different socioeconomic characteristics compared to west Nile region, Uganda. West Nile region is one of the refugee hosting regions, with Arua as the only city with high population and rapidly increasing chicken production to address the high food demand, yet the epidemiology and factors associated with MDR *E. coli* carriage on chicken farms are not well known. The human population dynamics and demand for animal-based protein especially chicken and chicken products could drive the emergency and spread of MDR pathogens within humans and animals in west Nile region. Therefore, the study sought to determine the prevalence, describe the AMR patterns and identify risk factors associated with MDR *E. coli* carriage on chicken farms. The results from this study can inform strategies to control the emergence and spread of AMR in poultry farming.

## Materials and methods

### Study area and setting

The study was conducted in Arua district located in west Nile sub-region in northern Uganda (3˚ 01' 28.80" N, 30˚ 54' 21.59" E). Due to the new city status, Arua's boundaries have increased from 10.5 Km$^2$ to 60 Km$^2$ and the population has grown seven-fold, from 60,000 to about 400,000 people [34]. The district is also currently providing shelter to over 200,000 refugees [34]. With the high influx of refugees from Southern Sudan and Democratic Republic of Congo, the demand for food and healthcare has increased [34]. The explosive population may modify the food production system into intensified production systems, which are drivers of AMR [30]. Over 85% of the livestock farmers in the district rear poultry for livelihood [4]. There has been reports on unrestricted open-market sale of veterinary antimicrobial drugs to livestock farmers especially poultry farmers in Ugand with Arua district not exception [35].

### Study design

A cross-sectional study was conducted among chicken rearing households with apparently healthy chickens (no cases of ill health during the study) from 6$^{th}$ April to 31$^{st}$ May 2023. The study included households keeping chicken of a flock size of at least 50 chickens for commercial and semi-commercial purposes, and had prior use of antibiotics. In a flock size of above 50, chickens are often kept under intensive management system associated with high amount of antibiotic use [30] and exposure to humans due to close proximity [32]. The chickens should have been intended for human consumption and had reached slaughter weight for broiler or were "spent" layers. Farms with apparently sick chicken and had administered any class of antibiotic(s) at any dose(s) within one week prior to the study were excluded. Additional data were obtained from the owners/managers of chicken farms on socio-demographic characteristics of farmers and the husbandry practices on the farms, using an observation checklist and a semi-structured questionnaire. The data collected using the observation checklist and the semi-structured questionnaire were used to determine the risk factors associated with MDR *E. coli* carriage on the chicken farms. Ten farmers randomly selected outside the study region were used to test the clarity, reliability, and validity of the questionnaire. Household members aged 18-years and above, and directly involved in the day to day management of the farm were interviewed. All respondents gave informed consent prior to sample collection or the interviews after explanation of the purpose of study to chicken farmers.

## Sampling procedure

A multistage sampling approach was used to reach the primary (farm) and secondary (chicken and human) sampling units of the study. First, 11 sub counties in Arua district were purposively selected with the help of the district veterinary office based on the farming characteristics. Secondly, following a proportion to size probability distribution, a total of 55 parishes were selected randomly from each sub-county based on the number of chicken farms in the subcounty. Thirdly, a total of 82 villages were randomly selected from the parishes based on the proportion to size probability distribution. All households were listed with support from the village chairperson and only those keeping chicken were eligible for sampling. A random number generation technique was used for the selection of households keeping chicken obtained from respective village chairpersons. Lists of sub-counties, parishes and villages were obtained from the district veterinary office. A minimum distance of 100 meters between farms was considered. At this distance the possibility of cross-contamination resulting from close proximity would be minimized. From each farm, 10 fresh faecal samples were picked from 10 imaginary squares in the farm unit and pooled in Amies transport media with charcoal to form a farm sample. On each farm, a household member (farmers) who fulfilled the study selection criteria was purposively selected to be interviewed.

## Sample size determination

A total of 158 chicken farms were included derived using the modified Kish Leslie sample size determination formula [Equation 1] [36], based on previous reports on the prevalence of MDR *E. coli* carriage of 88.4% [26], at a 95% confidence interval and 5% precision. Using Steve Bennett's formula of cluster sample size calculation [Equation 2] [37], a maximum of nine clusters (villages) were sufficient from each parish. The study considered a respondent from each farm, and a 0.2 level of homogeneity, and a design effect of 32.4 was obtained [Equation 3] [37].

**Equation 1: Sample size**, $N = (Z^2 PQ)/d^2$ [36], where N is the minimum sample size; Z was 1.96 (standard normal deviation at 95% confidence interval). P was the prevalence of carriage of MDR *E. coli* considered at 88.4% [26]; Q is 1-P, Q = 1−0.884; therefore, Q = 0.116; d was the maximum error allowed between the estimated prevalence; thus, d = 5% (95% confidence interval); $N = (Z^2 PQ)/d^2$; N = ([1.96]$^2$ x [0.884x0.116])/ [0.05]$^2$); N = 158.

**Equation 2: Number of clusters**, $C = p(1-p)D/d^2 b$ [37], where C was the number of clusters which we considered a village as a cluster, p was prevalence of carriage of MDR *E. coli* considered at 88.4% [26]; D was design effect and b was the expected number of respondent considered at 158 (a respondent per farm sampled). Design effect (D) was calculated using **equation 3**, $D = 1+(b-1)\delta$ [37], where $\delta$ was estimated at 0.2; D = 1+(158−1)0.2 = 32.4. Therefore, C = 0.884(1−0.884)32.4/0.05$^2$*158 = 8.41 ≈ 9. This sample size was enough for achieving the study outcomes at 80% power and 95% confidence interval.

## Data collection procedures

Study teams composed of one guide and an animal health officer, who identified, recruited and collected data from the farming households that fulfilled the study inclusion criteria. The study teams were trained on data collection and laboratory standard operating procedures prior to the study. A pre-tested semi-structured questionnaire was administered to the manager or owner of the farm with knowledge of the key activities around the farm so as to assess the risks associated with MDR *E. coli* carriage. Demographic characteristics of chicken farmers including location, sex (gender), age, marital status, education status, source of livelihood and awareness on AMR while chicken farms characteristics including chicken type, production

system and flock size were collected. Husbandry practices on the chicken farms collected included hand hygiene practices, presence of foot bath, changing room and staff working on other farms, litter disposal, management of farm manure, source of water, other livestock on farm, veterinary consultation, source of drugs, vaccination status, frequency of treatment, drug use status, decisions on chickens which were on medication and frequently used antibiotics on the farm. Fresh chicken faecal samples were collected from the farm unit with the oldest flock using sterile swab, and put in a sample collection tube containing Amies transportation media containing charcoal (Oxoid, UK). Samples were collected from the unit with oldest flock because they were considered to have reached their production cycle and were about to enter the food value chain. The samples were transported in Ziplocs and under cold chain (4 –< 10˚C) to the National Microbiology Reference Laboratory at the National Animal Disease Diagnostics and Epidemiology Center (NADDEC) in Entebbe within 72 hours of collection for laboratory analysis.

## Isolation and identification of *Escherichia coli*

A faecal sample was inoculated in sterile brain heart infusion (BHI) and incubated aerobically at 37˚C for 18–24 hours. A loopful of homogenised turbid suspension was streaked on Mac-Conkey agar with crystal violet (Oxoid, UK) and incubating aerobically at 37˚C for 18–24 hours. Bacterial colonies obtained were purified by sub-culturing on 5% sheep blood agar and incubated aerobically at 37˚C for 18–24 hours. The pure colonies were subjected to Gram staining and, oxidase test and then to standard identification biochemical tests that included Sulphide-Indole-Motility (SIM), Triple Sugar Iron (TSI), urease and citrate utilization according to standard operating procedures to identify *E. coli*. Isolates that were lactose fermenting on MacConkey, Gram negative rods and, oxidase negative, Sulphur, urea and citrate negative, but positive for indole and motility with overall acid reactions on TSI were presumptively identified as *E. coli*. All isolates presumptively identified as *E. coli*, were confirmed using BD Phoenix M50. The isolates were then re-sub-cultured on 5% blood agar to obtain fresh colonies of 18–24 hours old for antibiotic susceptibility testing (AST). An isolate of *E. coli* was chosen from each farm in the study for AST.

## Phenotypic antimicrobial susceptibility testing

Phenotypic antimicrobial susceptibility testing (AST) for *E. coli* isolates was conducted according to the standard operating procedure, using Kirby-Bauer disc diffusion method on Mueller-Hinton Agar method [38] and interpreted as either resistant, intermediate or susceptible according to clinical breakpoints as defined by the Clinical and Laboratory Standards Institute (CLSI) 2023 guidelines (version 33). Isolates with an intermediate susceptible result were considered resistant. The antimicrobials tested were commonly used antibiotics and those listed by WHO [39] shown in Table 1. Standard reference strains of *E. coli* (ATCC 25922) were used to quality control the antibiotic discs potency and Mueller Hinton Agar (MHA) performance characteristics. Multidrug resistance was defined as resistance to three or more classes of antibiotics [40]. A farm was defined as "positive" for a resistant *E. coli* if the randomly pick *E. coli* isolate from the farm was resistant to the antibiotics under study. A phenotypic AST was conducted on 158 *E. coli* isolates, with each isolate representing a chicken farm from the study.

## Statistical analysis

Data were entered into Microsoft Excel, cleaned and exported to STATA Statistical Package (version 16.0 Stata Corp LP, College Station, TX, USA) for analysis. The study employed univariate, bivariate and multivariable levels of analysis to assess factors associated with carriage

**Table 1. List of antimicrobials used for susceptibility testing (all antibiotic discs were from Oxoid, UK).**

| WHO CIA classification | Antimicrobial agent | Class | WHO-AWaRe | Disc Charges |
|---|---|---|---|---|
| Critically important | Ampicillin | Penicillin | Access | 10µg |
| | Gentamicin | Aminoglycoside | | 10 µg |
| | Ceftriaxone | Cephalosporin | Watch | 30 µg |
| | Ciprofloxacin | Fluoroquinolone | | 5 µg |
| | Imipenem | Carbapenem | | 10 µg |
| | Cefepime | Cephalosporin | | 30µg |
| Highly important | Chloramphenicol | Amphenicol | Access | 30 µg |
| | Cotrimoxazole | Diaminopyrimidine/sulphonamide | | 25 µg |
| | Tetracycline | Tetracycline | | 30 µg |

**WHO CIA** = World Health Organization's List of Critically Important Antimicrobials for Human Medicine, **WHO-AWaRe** = World Health Organization's Access, Watch, Reserve classification of Antibiotics for Human Medicine.

of MDR *E. coli* on chicken farms. Descriptive statistics were populated in one-way tabulations, frequencies and percentages were reported in tables and figures. Prevalence was defined as the total number of farms with MDR *E. coli* isolates out of the total number of chicken farms under consideration. Resistance of *E. coli* isolates were categorized as resistant or susceptible using the CLSI (version 33) guidelines. At bivariate analysis, the study assessed for the statistical association between covariates using the Pearson chi-square test. The study also employed a modified Poisson regression modelling approach at bivariate and multivariable analysis to generate prevalence ratios (PR) and the respective 95% confidence intervals (95% CI). Variables with p-value less than 0.25 and those deemed biologically important to the study from literature and expert opinion were included in multivariable analysis. At the multivariable level, a forward stepwise regression model approach was used where variables with P-value $<0.05$, were considered statistically significant and therefore associated with MDR. The variables, which formed the final model, were assessed for interaction by including an interaction term (each of the interaction terms was assessed for its statistical significance). Potential confounding between MDR *E. coli* (outcome variable) and independent variables that were considered in the final model were assessed and a covariate with a percentage change more than 10% was considered a confounder.

### Ethical considerations

Ethical approval was obtained from the Makerere University School of Public Health Ethics Committee on 4[th] April 2023 (Approval number: MakSPH-REC-182). Permission was sought from the Ministry of Agriculture, Animal Industry and Fisheries, Uganda and the district veterinary office prior to commencement of the study. A written consent was obtained from all the chicken farm owners before they and their farms were included in the survey.

### Results

### Farmer demographic and chicken farm characteristics surveyed in west Nile region of Uganda

Out of the 158 chicken farms investigated, over 52% (83/158) were located in the rural. Majority 60% (95/158) of the chicken farmers were females, 54% (85/158) aged 35 or more years, 87% (137/158) married and 47% (74/158) had attained tertiary education. In more than half, 59% (93/158), of households, poultry was a source of livelihood. Up to 65% (102/158) had kept chicken for utmost 5 years while 55% (85/158) were not aware of AMR. Less than half of the

**Table 2. Farmer demographic and chicken farm characteristics (n = 158) in west Nile region of Uganda.**

| Variable | Sub category | Frequency (n) | Percentage (%) |
|---|---|---|---|
| Subcounty | Urban | 32 | 20.2 |
| | Peri-urban | 43 | 27.2 |
| | Rural | 83 | 52.5 |
| Sex (gender) | Male | 63 | 39.9 |
| | Female | 95 | 60.1 |
| Age (years) | 18–24 | 15 | 9.5 |
| | 25–34 | 58 | 36.7 |
| | >35 | 85 | 53.8 |
| Marital status | Married | 137 | 86.7 |
| | Single | 21 | 13.3 |
| Education status | Informal | 14 | 8.9 |
| | Primary | 30 | 19.0 |
| | Secondary | 40 | 25.3 |
| | Tertiary | 74 | 46.8 |
| Poultry as source of livelihood | Yes | 93 | 58.9 |
| | No | 65 | 41.1 |
| Duration as poultry farmer (years) | < = 5 | 102 | 64.6 |
| | >5 | 56 | 35.4 |
| Farmer is aware of AMR | Yes | 71 | 45.5 |
| | No | 85 | 54.5 |
| Chicken type | Dual | 106 | 67.1 |
| | Meat (broiler) | 42 | 26.6 |
| | Egg (layer) | 10 | 6.3 |
| Production system | Free range | 19 | 12.0 |
| | Semi intensive | 68 | 43.1 |
| | Intensive | 71 | 44.9 |
| Flock size | Small scale (>50–100) | 107 | 67.7 |
| | Medium scale (101–1000) | 51 | 32.3 |

farms 45% (71/158) were kept under intensive production system and mostly 68% (107/158) in small scale (Table 2).

**Husbandry practices on the chicken farms surveyed in west Nile region of Uganda.** Out of the 158 chicken farms survey, majority of the farmers 71% (112) reported to have not been washing hands after handling chicken on the farm. Over 67% (107) of the farms had no footbath while 83% (131) and 84% (133) had no changing room and staff working on other farms respectively. Above half of the farms 58% (91) disposed off litter generated from the farm on monthly basis and over 70% used manure generated from litter on farm as fertilizer. Above 53% (84) of the farms used borehole as water source and 67% (105) had other livestock. Fifty eight percent (92) of the farmers administered drug doses as prescribed by qualified animal health worker while over 85% (135) sold their chicken while on medication. Tetracycline (83.6%), sulfadimidine (32.9%) and cotrimoxazole (9.2%) were the frequently used antibiotics on chicken farms by the farmers (Table 3).

## Prevalence of multidrug resistant *E. coli* species and resistance patterns on chicken farms

Out of the 158 farms where fresh feacal samples were collected, all the samples tested positive to *E. coli* with a prevalence of isolation of 100%. After a susceptibility testing, 151 *E. coli* isolates

**Table 3. Husbandry practices on the chicken farms surveyed in west Nile region of Uganda.**

| Variable | Sub category | Frequency (n) | Percentage (%) |
|---|---|---|---|
| Washed hands after handling chicken on farm | Yes | 46 | 29.1 |
| | No | 112 | 70.9 |
| Availability of foot bath on farm | Yes | 51 | 32.3 |
| | No | 107 | 67.7 |
| Changing room present on farm | Yes | 27 | 17.1 |
| | No | 131 | 82.9 |
| Staff working on other farms | Yes | 25 | 15.8 |
| | No | 133 | 84.2 |
| Frequency of litter disposal | Everyday | 29 | 12.7 |
| | Weekly | 36 | 22.8 |
| | Monthly | 91 | 57.6 |
| | When stock sold off | 11 | 7.0 |
| Management of manure on the farm | Bag and sell | 18 | 11.4 |
| | In pit | 29 | 18.4 |
| | Use as fertilizer | 111 | 70.3 |
| Water source | Borehole | 84 | 53.2 |
| | Tap | 64 | 40.5 |
| | Well | 10 | 6.3 |
| Other livestock | Yes | 105 | 66.5 |
| | No | 53 | 33.5 |
| Veterinary consultation | Veterinary officer | 133 | 84.2 |
| | Fellow farmer | 25 | 15.8 |
| Drug provider on the farm | Government veterinarian | 49 | 31.0 |
| | Pharmacy | 86 | 54.4 |
| | Private veterinarian | 23 | 14.6 |
| Vaccinated chicken | Yes | 72 | 45.6 |
| | No | 86 | 54.4 |
| Frequency of treating chicken | When there is disease | 63 | 39.9 |
| | When production reduces | 37 | 23.4 |
| | When they are sick | 58 | 36.7 |
| Drug doses administered as prescribed | Yes | 92 | 58.2 |
| | No | 66 | 41.8 |
| Sell chicken on medication | Yes | 135 | 85.4 |
| | No | 23 | 14.6 |
| Frequently used antibiotic | Tetracycline | 127 | 83.6 |
| | Sulfadimidine | 50 | 32.9 |
| | Cotrimoxazole | 14 | 9.2 |
| | Enrolfloxacin | 10 | 6.6 |

were resistant to at least a class of antibiotic with overall resistance prevalence of 95.6% (95% CI:91.1–98.2). Moreover, 99/158 showed a resistance to at least three antibiotics for a multi-drug resistance prevalence of 62.7% (95% CI: 55.0–70.3).

Overall, 158 *E. coli* isolates from the 158 chicken farms sampled were tested against the antibiotics included in the study to determine their resistance patterns. On average, among the common WHO CIA and WHO AWaRe categories for human medicine, higher resistance (50%) was observed among WHO Access and Highly Important antimicrobials (Access-HIA), followed by Critically Important category (Access-CIA), 48%. Relatively lower resistance

**Table 4. Patterns of chicken farms *E. coli* resistance against antimicrobials critically important in human medicine.**

| WHO CIA classification | Antimicrobial agent | WHO-AWaRe | Number of resistant *E. coli* isolates | Percentage (%) | 95% CI |
|---|---|---|---|---|---|
| Highly important | Chloramphenicol | Access | 33 | 20.9 | 15.2–28.0 |
| | Cotrimoxazole | | 88 | 55.7 | 47.8–63.3 |
| | Tetracycline | | 115 | 72.8 | 65.2–79.2 |
| Critically important | Ampicillin | Access | 126 | 79.8 | 72.7–85.4 |
| | Gentamicin | | 26 | 16.5 | 11.4–23.2 |
| | Ceftriaxone | Watch | 13 | 8.2 | 4.8–13.7 |
| | Ciprofloxacin | | 60 | 38 | 30.7–45.9 |
| | Imipenem | | 15 | 9.5 | 5.8–15.2 |
| | Cefepime | | 11 | 7 | 3.9–12.3 |

**WHO CIA** = World Health Organization's List of Critically Important Antimicrobials for Human Medicine, **WHO-AWaRe** = World Health Organization's Access, Watch, Reserve classification of Antibiotics for Human Medicine.

(16%) was observed among the Watch and Critically Important Antimicrobials category (Watch-CIA). Among the Access-HIA category, highest resistance was observed against tetracycline, 72.8% (95% CI 65.2–79.2) while among the Access-CIA, resistance was highest against ampicillin 79.8% (95% CI 72.7–85.4). Among the Watch-CIA category, highest resistance was observed against ciprofloxacin, 38% (95% CI 30.7–45.9) and lowest against cefepime, 7% (95% CI 3.9–12.3) (Table 4).

The multidrug resistance phenotypic patterns of all the *E. coli* isolates (n = 99) are shown in Table 5. Among the MDR *E. coli* isolates, 16 MDR phenotypic patterns were exhibited with AMP, CHL, CIP being the predominant 23.2% (23/99). Furthermore, out of the 99 MDR *E. coli* isolates, an isolate was resistant to 6/9 antibiotic classes in the study (Table 5).

**Table 5. Phenotypic multidrug resistance profile of *E. coli* (n = 99) isolated on chicken farms west Nile region of Uganda to tested antimicrobials.**

| Number of antibiotics class | Phenotypic resistance pattern | Number of *E. coli* isolates | Percentage (%) |
|---|---|---|---|
| **Three** | AMP, CHL, CIP | 23 | 23.2 |
| | AMP, CRO, CHL | 13 | 13.1 |
| | AMP, CIP, GEN | 11 | 11.1 |
| | AMP, IMP, SXT | 7 | 7.1 |
| | AMP, IMP, TCY | 7 | 7.1 |
| | AMP, FEP, SXT | 5 | 5.1 |
| | AMP, FEP, TCY | 5 | 5.1 |
| **Four** | AMP, IMP, SXT, TCY | 7 | 7.1 |
| | AMP, CIP, GEN, TCY | 6 | 6.1 |
| | AMP, CHL, CIP, GEN | 4 | 4.0 |
| | AMP, CIP, GEN, SXT | 2 | 2.0 |
| | AMP, CRO, CHL, CIP | 1 | 1.0 |
| **Five** | AMP, CHL, CIP, GEN, TCY | 4 | 4.0 |
| | AMP, CIP, GEN, TCY, SXT | 2 | 2.0 |
| | AMP, CRO, CHL, CIP, SXT | 1 | 1.0 |
| **Six** | AMP, CRO, CHL, CIP, SXT, TCY | 1 | 1.0 |

AMP = ampicillin, CHL = chloramphenicol, CIP = ciprofloxacin, CRO = ceftriaxone, FEP = cefepime, GEN = gentamycin, IMP = imipenem, SXT = cotrimoxazole, TCY = tetracycline.

### Public health risk factors associated with carriage and spread of MDR *E. coli* on chicken farms of farming households

Factors independently associated with carriage and spread of MDR *E. coli* included gender, absence of footbath, education level and antimicrobial dose regimen. At bivariate analysis level, MDR varied significantly by sex (gender) of the farmer (p = 0.004); being higher for females than males managing farms (68.7% vs 31.3% respectively). Additionally, the prevalence of MDR significantly varied by farmer's level of education (p = 0.017) and was substantially higher (64%) of MDR on farms without footbath (Table 6). At multivariable analysis after adjusting for factors included in the model, chicken farms managed by males had a lower prevalence of MDR *E. coli* (APR = 0.72, 95% CI: 0.55–0.93) compared to those managed by females. The prevalence of MDR *E. coli* was significantly (p = 0.002) higher among chicken farms that had no footbath (APR = 1.48, 95% CI: 1.16–1.88). Farms managed by farmers with education level higher than primary had a significantly lower likelihood of MDR *E. coli* (Table 6): secondary (APR = 0.64, 95% CI: 0.46–0.88) and tertiary education level (APR = 0.60, 95% CI: 0.47–0.75). Furthermore, farmers who administered the drug doses as prescribed were less likely to have farms with MDR *E. coli* (APR = 0.76, p = 0.022) (Table 6).

After assessing for effect modification, the study found a significant association between the prevalence of MDR *E. coli*, education level of the farmer and presence of footbath on farms. The farmers who attained tertiary education level but had no footbath on farms were two times more likely to have farms with MDR *E. coli* (APR = 1.74, p = 0.01). Potential confounding between MDR *E. coli* (outcome variable) and independent variables that were considered in the final model were also assessed, however, no potential confounding existed between the variables.

## Discussion

The study found a high prevalence of MDR *E. coli* (62.7%) on chicken farms. High resistance was observed against ampicillin, tetracycline, cotrimoxazole, and ciprofloxacin. Farms managed by female gender compared with their male gender had significantly higher prevalence of MDR *E. coli* (68.7% vs 31.3%) while significantly varied with farmers' level of education. Presence of footbath, secondary education or higher, and adherence to recommended antibiotic doses were protective against *E. coli* resistance while female managed farms and education with no footbath posed a heightened risk. There is need for context-specific collaborative strategies, to address knowledge on biosecurity on farms, and prudent use of antimicrobials among chicken farming communities, which considers gender dimensions to safeguard both animal and human health.

Multidrug resistant *E. coli* on chicken farms poses a significant public health threat because of *E. coli*'s tendency to disseminate AMR genes to intestinal bacteria in humans [5, 13]. The MDR *E. coli* prevalence in this study was higher than the prevalence observed in western Uganda (37%) [33] and the global average (40%) [41], implying a high burden of MDR *E. coli* on the Arua district farms. However, the results were similar to findings from a Vietnam study (63.3%) [42], but lower than the prevalence of 71% in Nepal [43]. According to a 2022 systematic review in Sub-Saharan Africa, the prevalence of MDR *E. coli* ranged from 98.7% in Uganda to 100% in Ghana [44]. The differences in MDR *E. coli* prevalence could be due to differences in husbandry systems, sample types and interpretation of antimicrobial susceptibility. The study in Nepal used cecum samples while the other study in Uganda used layer chickens under a deep litter intensive husbandry system, compared to on-farm fresh faecal samples collected from all chicken husbandry systems in this study. Furthermore, the study in Nepal used the minimum inhibitory concentration (MIC) CLSI guideline while our study used Kirby-

**Table 6. Prevalence of MDR *E. coli* and risk factors associated with carriage and spread of antimicrobial on chicken farms in west Nile region.**

| Factor | Multidrug resistance | | Bivariate | | Multivariable | |
|---|---|---|---|---|---|---|
| | Yes n (%) = 99(62.7) | No n (%) = 59(37.3) | UPR (95% CI) | Chi-square | APR (95% CI) | p-value |
| **Sex** | | | | 0.004* | | |
| Female | 68(68.7) | 27(45.8) | Ref | | Ref | |
| Male | 31(31.3) | 32(54.2) | 0.69(0.52–0.91) | | 0.72(0.55–0.93) | 0.012* |
| **Production system** | | | | 0.693 | | |
| Free range | 12(12.1) | 7(11.9) | Ref | | | |
| Semi intensive | 45(45.5) | 23(39.0) | 0.93(0.63–1.39) | | - | - |
| Intensive | 42(42.4) | 29(49.2) | 1.05(0.71–1.54) | | - | - |
| **Flock size** | | | | 0.155 | | |
| Small scale (>50–100) | 63(63.6) | 44(74.6) | Ref | | | |
| Medium scale (>101) | 36(36.4) | 15(25.4) | 1.20(0.94–1.52) | | - | - |
| **Foot bath** | | | | 0.155 | | |
| Yes | 36(36.4) | 15(25.4) | Ref | | Ref | |
| No | 63(63.6) | 44(74.6) | 1.20(0.94–1.52) | | 1.48(1.16–1.88) | 0.002* |
| **Marital Status** | | | | 0.063 | | |
| Married | 82(82.8) | 55(93.2) | Ref | | | |
| Single | 17(17.2) | 4(6.8) | 1.35(1.05–1.74) | | - | - |
| **Farmer's education level** | | | | 0.017* | | |
| Informal | 14(14.1) | 0 | Ref | | Ref | |
| Primary | 20(20.2) | 10(16.9) | 0.67(0.52–0.86) | | 0.74(0.55–1.02) | 0.064 |
| Secondary | 22(22.2) | 18(30.5) | 0.55(0.42–0.73) | | 0.64(0.46–0.88) | 0.007* |
| Tertiary | 43(43.4) | 31(52.5) | 0.58(0.48–0.71) | | 0.60(0.47–0.75) | <0.001* |
| **Drug doses administered as prescribed** | | | | 0.060 | | |
| No | 47(47.5) | 19(32.2) | Ref | | Ref | |
| Yes | 52(52.5) | 40(67.8) | 0.79(0.63–1.01) | | 0.76(0.59–0.96) | 0.022* |
| **Drug provider** | | | | 0.084 | | |
| Government veterinarian | 30(30.3) | 19(32.2) | Ref | | Ref | |
| Pharmacy | 59(59.6) | 27(45.8) | 1.12(0.86–1.46) | | 1.07(0.79–1.44) | 0.678 |
| Private veterinarian | 10(10.1) | 13(22.0) | 0.71(0.42–1.19) | | 0.62(0.38–1.01) | 0.054 |
| **Frequency of treating chicken** | | | | 0.277 | | |
| When there is disease | 37(37.4) | 26(44.1) | Ref | | Ref | |
| When there is reduced | 21(21.2) | 16(27.1) | 0.97(0.68–1.37) | | 0.75(0.55–1.03) | 0.072 |
| When they are sick | 41(41.4) | 17(28.8) | 1.20(0.92–1.57) | | 1.20(0.93–1.54) | 0.153 |
| **Sell chicken on medication** | | | | 0.094 | | |
| No | 18(18.2) | 5(8.5) | Ref | | | |
| Yes | 81(81.8) | 54(91.5) | 0.77(0.59–0.99) | | - | - |

*Values that were significant at bivariate and multivariable analysis.

**UPR** = Unadjusted Prevalence Ratio; **APR** = Adjusted Prevalence Ratio; Adjusted for gender, production system, flock size, footbath, marital status, farmer's education, dose, drug provider, frequency of treating chicken and sell of chicken on medication.

Bauer disc diffusion on Mueller-Hinton agar method to interpret the antimicrobial suscepti-bility. Unlike MIC method which is quantitative, disc diffusion provides a qualitative AST result [45]. These high prevalence of MDR *E. coli* on chicken farms could reveal the burden in animal health. The MDR *E. coli* on chicken farms could spillover and amplified among human population and environment [32]. Infections by MDR *E. coli* in humans and animals, certainly drive the cost of treatment high, morbidity and mortality in both animals and humans

compromising quality of health outcomes and economic growth [1] thus, posing significant threat to public health.

There were significant differences in prevalence of MDR *E. coli* by gender and education level of the farmer. Differences in MDR *E. coli* prevalence by gender and education level have been observed in other studies [46, 47]. Knowledge and gender influence farmers' decision on when and how to use antimicrobials [46, 47], hence education targeting gender dimensions may be essential to promote responsible antimicrobial use practices and mitigating the emergency and spread of MDR *E. coli* on chicken farms, to prevent potential spillover into human population. The study found a low prevalence of MDR *E. coli* on farms managed by males compared to those managed by females, consistent with other similar studies [46, 48]. The low prevalence of MDR *E. coli* for farms managed by males could be attributed to the gender-based socioeconomic differences [3, 49]. Males compared to females counterparts have higher access to information and financial resources to seek healthcare in a household including veterinary services [3]. Limited access to information and financial resources influences decision on seeking for veterinary services and prudent antimicrobial use, thus promoting irrational antimicrobial use which drives emergency and spread of resistant pathogens including MDR *E. coli* on farms [46]. In contrast, a study conducted in Zambia reported females being more aware about antimicrobial uses and AMR, and they had good health seeking behaviour compared to their male counterparts [50]. Good health seeking behaviour is associated with low prevalence of AMR and MDR [50, 51]. The observed differences in these studies could be attributed to the variation in the socioeconomic practices of the study populations [51]. The high prevalence of on farm MDR *E. coli* reported in our study could also be attributed to the observed differences in education status of the farmers. The study found significant difference in prevalence of MDR *E. coli* by education level of the farmer, where almost all the chicken farms managed by farmers who did not attained any formal education status had MDR *E. coli*. The high prevalence of MDR *E. coli* on female managed farms especially with low level education reported in this study indicates that farmers' gender differences on prevalence of AMR on farms are greatly contextual and intersect with other sociodemographic factors, particularly education status [51]. Socioeconomic status and gender plays a critical role in health promotion of the family and community [52, 53]. Chicken contributes to nutrition security directly and indirectly through the sale and home consumption of chickens and eggs at both the household and community levels [54]. Therefore, the observed high prevalence of MDR *E. coli*, particularly on farms managed by females, raises concerns about the potential negative effects on maternal and child nutrition, as it could lead to increased nutrition insecurity, compromised health outcomes, and hindered economic growth [1]. Consequently, these can have a significant negative public health impact.

There was high resistance against ampicillin (Access-CIA), and tetracycline and cotrimoxazole (Access-HIA). Higher ampicillin resistance has been observed in Cameroon (77%) but a lower resistance has been observed in Tanzania (52.3%) [55]. Similar results were reported in Nepal (66%) [43], Vietnam (21.0%– 73.3%) [42], Cameroon (58.4%) [12] and Tanzania (54%) [56]. This situation may result from inadequate veterinary healthcare systems, which promotes irrational use of antimicrobials and drives AMR emergence and spread [10, 57, 58]. Gentamycin (Access-CIA) resistance was higher in Cameroon (38%) [59] compared to our study finding. Lower resistance was observed against the Watch-CIA with highest resistance being against ciprofloxacin. The lower *E. coli* resistance to Watch-CIA: cefepime and ceftriaxone (7%) and imipenem (11.6%) was similar to retrospective study done in the Central Diagnostic Laboratory (CDL) at Makerere University in Uganda [26]. These drugs are uncommon for use in livestock in Uganda therefore limiting bacterial exposures. However, the observed resistance is an indication of a raising resistance against drugs considered as last-resort in human

medicine which is worrisome for public health. The high resistance to ciprofloxacin could be attributed to the high level of use of enrofloxacin in poultry [60]. The high *E. coli* resistance to Access-HIA in this study was similar to that in Tanzania: chloramphenicol (34.2%), cotrimoxazole (56.7%) and tetracycline (68.4%) [55]. However, tetracycline resistance in this study was lower than that reported in Bangladesh (100%) [61], Malaysia (91.2%) [62] and Nepal (86%) [43]. In contrast, lower resistance against chloramphenicol were also reported in Kenya (2%) [63], Zambia (8.8%) [48] and Canada (17%) [64]. This could be attributed to differences in interpretation of resistance, sample sources and variation in implementation of drug related and AMR mitigation policies [32, 65]. The observed resistance patterns to these critically important antimicrobial agents in human Medicine in this present study is worrisome for public health since it may limit treatment options compromising health outcomes [66].

Sex and education of a farmer, absence of footbath on the farm and following dose regimen were associated with MDR *E. coli* consistent with other studies [10, 30, 57, 67]. Chicken farms managed by female gender heightened risk of MDR *E. coli* consistent to reports in Zambia [48] and Pakistan [68]. Several factors could be attributed to the observed heightened risk of MDR *E. coli* on farms managed by females, which includes gender roles in managing farms [17] and limited access to resources and information on good husbandry practices [51, 53, 69]. Poor husbandry practices on farms has been shown to influence the emergence and spread of AMR [30]. The observed high likelihood of female managed farms having high prevalence of carriage of MDR *E. coli* could pose a significant public health threat because of the socio-cultural roles of female gender (women and girls) at the household and community levels which includes taking care of children, household WASH activities [18]. Thus these roles played by female gender, coupled with the observed heightened risk of MDR *E. coli* on chicken farms exposes females to resistant pathogens, which could lead to spillover and amplification into the human population, especially among chicken farming communities due to these interactions [14, 17]. The negative public health impact could potentially affect these chicken farming communities more especially mothers and children who are more vulnerable to resistant zoonotic pathogens due to limited access to WASH facilities [16, 18, 70].

In this study, farms without footbath were more likely to have a high prevalence of MDR *E. coli* similar to studies in Sub-Saharan Africa [32, 71]. Presence of a footbath on farms is one of the basic biosecurity measures. The poor biocontainment practices, and the close interactions between humans and animals as observed among the farming households in this study, may pose a potential pathway for spread of resistant pathogens between human-animal interface [32, 72]. The observed high likelihood of farms without footbath having high risk of carriage of MDR *E.coli* shows the potential for environmental contamination, thus limiting access to clean WASH. Lack of access to clean water for both humans and animals have a significant public health threat as it provides pathway for transmission of resistant zoonotic pathogens between human-animal and environment interface [72]. Inadequate sanitation exposes household members especially children to chicken faecal matter thus exposure to antibiotic resistant pathogens [18]. Infections by these zoonotic pathogens could lead to high morbidity and mortality among humans and animals, and food insecurity among the chicken farming communities [1]. A farmer attaining at least a secondary education was associated with a lower risk of MDR *E. coli* on the chicken farms similar to finding of a study in Bangladesh [10]. This could be attributed to enhanced knowledge on AMR and antimicrobial use. However, this study found a significant interaction between education and absence of footbath on chicken farms. Importantly, farmers who attained a tertiary education level but had no footbath on their farms were more likely to have farms with MDR *E. coli* on their farms. This means education without footbath on chicken farms could not protect against occurrence of MDR *E. coli* on farms [14, 32]. This study indicates that public health capacity building interventions should

be highly contextual since education alone may not be enough to contain emergence and spread of MDR *E. coli* on chicken farms.

There were some methodological limitations in this study, including using responses from farmers to test for association with MDR *E. coli* which could have introduced a recall bias. The study did not also look at AMR transmission from chicken to human. However, this limitation was mitigated by using a pretested and validated—for face, content and criterion—semi-structured questionnaire to determine risk factors associated with MDR *E. coli* on chicken farms. Secondly, the study did not perform molecular characterization of all the MDR isolates. However, the phenotypic analysis conducted was based on the internationally recognized and standardized CLSI protocol and guidelines, thus reflecting the real magnitude and pattern of AMR in the study setting.

This study provides data on the prevalence and risk factors associated with carriage of MDR *E. coli* on chicken farms which can be used to guide evidence-based veterinary and public health policies for the containment of emergence and spread of AMR between human, animal and environment interfaces.

## Conclusions

There was a high prevalence of MDR *E. coli* on chicken farms, which varied by sex and education level of the farmers. Resistance to ampicillin, tetracycline and cotrimoxazole among the Access antibiotic class and ciprofloxacin among the Watch antibiotic class was common. Footbath, adherence to recommended antibiotic doses and secondary education or higher posed a low risk of *E. coli* resistance, while education with no footbath posed a heightened risk. Antimicrobial stewardship and infection prevention strategies are a paramount requirement on chicken farms. Further studies should consider environmental and human samples from the chicken farmers to compare the resistance patterns in these communities. Genotypic and molecular studies should also be performed on isolates in order to understand the transmission of resistomes across the human, animal, and environment interface.

## Supporting information

**S1 File. Questionaire.**
(DOCX)

**S2 File. Data set.** Values used for the data analysis.
(XLS)

**S3 File. Standard operating procedure for collection of faecal sample on chicken farm.**
(DOCX)

## Acknowledgments

The authors would like to appreciate the respondents who participated in this study. The authors also appreciated the management of the National Animal Disease Diagnostic and Epidemiology Center (NADDEC) of the Ministry of Agriculture, Animal Industry and Fisheries, Uganda (MAAIF) for allowing the study to be conducted successfully. The authors further appreciated the support provided by the Arua District Veterinary Office during mobilization of farmers, farm identification and sample collection. The authors also acknowledged the intellectual contributions of Ms. Vivian Twemanye, Jonathan Kabazzi, Christopher Majwega and Obote Daniel towards the success of this research.

## Author Contributions

**Conceptualization:** Ceaser Adibaku Nyolimati, Jonathan Mayito, Richard Walwema, David Musoke, Christopher Garimoi Orach.

**Data curation:** Ceaser Adibaku Nyolimati, Emmanuel Obuya.

**Formal analysis:** Ceaser Adibaku Nyolimati, Emmanuel Obuya.

**Funding acquisition:** Jonathan Mayito, Richard Walwema, Francis Kakooza.

**Investigation:** Ceaser Adibaku Nyolimati, Atim Stella Acaye, Emmanuel Isingoma, Daniel Kibombo.

**Methodology:** Ceaser Adibaku Nyolimati, Daniel Kibombo, Richard Walwema, David Musoke, Christopher Garimoi Orach.

**Project administration:** Ceaser Adibaku Nyolimati, Jonathan Mayito, D. M. Byonanebye, Richard Walwema, Francis Kakooza.

**Resources:** Ceaser Adibaku Nyolimati, Jonathan Mayito, Atim Stella Acaye, Emmanuel Isingoma, D. M. Byonanebye, Richard Walwema, Francis Kakooza.

**Software:** Ceaser Adibaku Nyolimati, Emmanuel Obuya.

**Supervision:** Ceaser Adibaku Nyolimati, Jonathan Mayito, Atim Stella Acaye, Emmanuel Isingoma, Richard Walwema, David Musoke, Christopher Garimoi Orach, Francis Kakooza.

**Validation:** Ceaser Adibaku Nyolimati, David Musoke, Christopher Garimoi Orach.

**Visualization:** Ceaser Adibaku Nyolimati, Emmanuel Obuya.

**Writing – original draft:** Ceaser Adibaku Nyolimati, Daniel Kibombo, David Musoke, Christopher Garimoi Orach.

**Writing – review & editing:** Ceaser Adibaku Nyolimati, Jonathan Mayito, Atim Stella Acaye, Emmanuel Isingoma, Daniel Kibombo, D. M. Byonanebye, David Musoke, Christopher Garimoi Orach.

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
