## [Decision Letter · Decision Letter 0]

20 Aug 2024

PGPH-D-24-01295

Prevalence and factors associated with multidrug resistant Escherichia coli carriage on chicken farms in west Nile region in Uganda: A cross-sectional survey

Dear Dr. Nyolimati,

Thank you for submitting your manuscript to PLOS Global Public Health. After careful consideration, we feel that it has merit but does not fully meet PLOS Global Public Health’s publication criteria as it currently stands. Therefore, we invite you to submit a revised version of the manuscript that addresses the points raised during the review process.

Thank you for submitting your manuscript to PLOS Global Public Health. Your manuscript has been reviewed by two referees who both found your manuscript of interest, whose feedback can be found below. Please revise your manuscript according to the reviewers' suggestions and submit the updated version along with a response letter detailing the changes made. We look forward to receiving your revised manuscript.

While your study was praised for its robust methodology, clear statistical analysis, and significant public health implications, please provide additional details on the stool sample collection process, including who collected the samples and the protocols used to ensure reproducibility.

We look forward to receiving your revised manuscript.

Kind regards,

Ben Pascoe

Academic Editor

Journal Requirements:

1. We do not publish any copyright or trademark symbols that usually accompany proprietary names, eg (R), (C), or TM  (e.g. next to drug or reagent names). Please remove all instances of trademark/copyright symbols throughout the text, including ® on page 9.

Additional Editor Comments (if provided):

Reviewers' comments:

Reviewer's Responses to Questions

**Comments to the Author**

1. Does this manuscript meet PLOS Global Public Health’s publication criteria? Is the manuscript technically sound, and do the data support the conclusions? The manuscript must describe methodologically and ethically rigorous research with conclusions that are appropriately drawn based on the data presented.

Reviewer #1: Yes

Reviewer #2: Yes

2. Has the statistical analysis been performed appropriately and rigorously?

Reviewer #1: Yes

Reviewer #2: Yes

3. Have the authors made all data underlying the findings in their manuscript fully available (please refer to the Data Availability Statement at the start of the manuscript PDF file)?

Reviewer #1: Yes

Reviewer #2: Yes

4. Is the manuscript presented in an intelligible fashion and written in standard English?

Reviewer #1: Yes

Reviewer #2: Yes

5. Review Comments to the Author

Reviewer #1: Title:

Prevalence and factors associated with Multidrug Resistant Escherichia coli carriage on chicken farms in west Nile region in Uganda: A cross-sectional survey.

Abstract

Line 18-19: “A total of 158 E. coli strains were isolated from 158 chicken farms”. What is the study population (chicken farms or E. coli strains)? 158 strains for 158 chicken farms? Does it mean each chicken with its strain?

Introduction

Line 41-42: AMR reported in animals or in humans? As the authors have studied AMR in poultry, it is important to specify if this sentence is related to human or animals or both.

Line 48: predisposition of emergence AMR to humans or animals?

Lines 52-54: Is there any existing policy for AMR control in Uganda? If yes, better to start this before speaking about laxity of policy implementation.

Line 59: Fix the grammar issue in this sentence.

Line 61: There is repetition of “the”

Line 55-62: “Gender determines the division of labour in many contexts especially in Low- and Middle-Income Countries (LMICs). In most of the farming communities in Sub Saharan Africa (SSA), female gender particularly, women take care of sick animals not destined for slaughter while the male gender owns the farms. Compared to the male gender counterparts, females gender also plays critical roles in household activities and taking care of children including household water, sanitation and hygiene (WASH) activities. These roles increases their exposure to and spread of resistant zoonotic pathogens including the drug resistant ones“ => is this paragraph relevant to this study? Are the authors studying the level of AMR between males and females?

Line 66: recent past? => it is better to be precise about a period (recent past years, recent past months, ect.).

Line 70: The sentence “… distributed among humans, animals and environment” should read “… distributed among humans, animals, and environment”.

Methods

Line 111-117: What is the study population here (poultry or humans (farmers))? What is the purpose of the questionnaire for this study? It is important to be clear about that. How the households were selected to be included in the study?

Lines 118-131: Sampling: this is good details about selection of the chickens. So, what about the farmers who attended the survey questionnaire?

Lines 157-159: Are these demographic characteristics for chickens or for farmers?

Lines 183-184: The following sentence looks like a result and should be in the results section: “All the farms sampled tested positive for E. coli and an isolate was chosen from each of the 158 farms for AST”.

Results

Table 2: merge some rows for the column of variables to match the sub-categories.

Table 3: merge some rows for the column of variables to match the sub-categories.

Fig 1: It will be good to add labels to your figure. As it is, it looks like all the 3 colors are equally distributed.

Line 292-293: The sentence “being higher on female managed farms than male managed farms (68.7% vs 31.3% respectively)” should read “being higher for females than males managing farms (68.7% vs 31.3% respectively)”.

Discussions

Lines 321-329: This paragraph is a repetition of the results. The authors can think if it necessary to keep it here. Otherwise, I would suggest to remove it as it is already stated in the results. And in the results, think about rephrasing the sentence about gender. As it is written, it looks like the MDR was studied for people rather than for chickens.

Line 356: The sentence: “The low prevalence of MDR E. coli on male managed farms …” should read “The low prevalence of MDR E. coli for farms managed by males …”

Line 370: Please think about rephrasing this sentence: “… prevalence of MDR E. coli among female gender …”. What you mean by female gender here? (human or animal?). You are studying MDR in chicken, not in human. Go through the whole text and be precise about this.

Lines 375-376: “the observed high prevalence of MDR E. coli especially on farms managed by females could impact negatively on maternal and child nutrition”. => this needs further details. It is still not clear where it comes from.

Line 387: The sentence “… in Makerere University…” should read “… at Makerere University …”

Lines 400-419: This paragraph is a repetition of what you stated above. Think about removing it or shorten it to essential information.

Line 445: Be sure you have stated clearly in the methods that the analysis was phenotypic rather than genotypic.

Add in the study limitation that the study did not look at AMR transmission from chicken to human and that this can be explored in future researches.

Conclusion

Line 455: “… protected against E. coli resistance …” => this is an over claim. May be say “… reduce E. coli resistance …”.

General comments:

This is a very good study that add value to the literature. Authors can consider the above comments to improve the paper.

Reviewer #2: The article meets the criteria for PLOS Global public health journal publication and has a technically sound manuscript with conclusions that are supported by data presented in the quantitative tables. The cross sectional design for the primary research question posed projects an ethically and methodologically sound design with mentioned limitations and weaknesses.

This is further elaborated by the drawn conclusions which even with the mentioned recall bias in the design, not only answers the question posed but go on to identify factors associated with and potentially propagating multi drug resistance among chicken farms in Uganda. Despite the mentioned practical hindrances to implementing antibiotic resistance awareness among female Uganda chicken farmers, the results and conclusion clearly show and advocates for such a vital public health intervention particularly in the developing nation. The conclusions also supports this notion.

An astounding statistical analysis of the data is seen through the sub-categorization of data captured from the snapshot survey and responses from the questionnaire. The mathematical formulas used are clearly shown and applied. Furthermore, the manuscript is presented in good palatable english.

A recommendation would be to provide details on the stool sample collection protocols eg who collected (qualified or not), standard operating protocols eg what measures were in place to ensure reproducibility of the sample collection process.

6. PLOS authors have the option to publish the peer review history of their article (what does this mean?). If published, this will include your full peer review and any attached files.

**Do you want your identity to be public for this peer review?** For information about this choice, including consent withdrawal, please see our Privacy Policy.

Reviewer #1: No

Reviewer #2: No

---

## [Decision Letter · Decision Letter 1]

26 Nov 2024

PGPH-D-24-01295R1

Prevalence and factors associated with multidrug resistant Escherichia coli carriage on chicken farms in west Nile region in Uganda: A cross-sectional survey

Dear Dr. Nyolimati,

Thank you for submitting your manuscript to PLOS Global Public Health. After careful consideration, we feel that it has merit but does not fully meet PLOS Global Public Health’s publication criteria as it currently stands. Therefore, we invite you to submit a revised version of the manuscript that addresses the points raised during the review process.

We look forward to receiving your revised manuscript.

Kind regards,

Ben Pascoe

Academic Editor

Journal Requirements:

Additional Editor Comments (if provided):

Based on the feedback from our reviewers, I am pleased to inform you that your manuscript has been evaluated positively and is suitable for publication following minor revisions.

The reviewers found your study to be a valuable contribution to the literature, addressing an important public health issue in antibiotic resistance. The study's design, conclusions, and statistical analyses were commended as sound and robust. The reviewers appreciated the clarity of your manuscript and the relevance of your findings, particularly in advocating for antibiotic resistance awareness interventions among female chicken farmers in Uganda.

To proceed with publication, please address the following minor revisions:

Methods: Clarify that the analysis conducted was phenotypic rather than genotypic.

Results: Adjust Figure 1 and present it as Table 4 for consistency.

Methods (lines 123–124 and 140–141): Remove the repetition of the sentence describing the selection criteria for interviewees.

Reviewers' comments:

Reviewer's Responses to Questions

**Comments to the Author**

1. If the authors have adequately addressed your comments raised in a previous round of review and you feel that this manuscript is now acceptable for publication, you may indicate that here to bypass the “Comments to the Author” section, enter your conflict of interest statement in the “Confidential to Editor” section, and submit your "Accept" recommendation.

Reviewer #1: (No Response)

Reviewer #2: All comments have been addressed

2. Does this manuscript meet PLOS Global Public Health’s publication criteria? Is the manuscript technically sound, and do the data support the conclusions? The manuscript must describe methodologically and ethically rigorous research with conclusions that are appropriately drawn based on the data presented.

Reviewer #1: Yes

Reviewer #2: Yes

3. Has the statistical analysis been performed appropriately and rigorously?

Reviewer #1: Yes

Reviewer #2: Yes

4. Have the authors made all data underlying the findings in their manuscript fully available (please refer to the Data Availability Statement at the start of the manuscript PDF file)?

Reviewer #1: Yes

Reviewer #2: Yes

5. Is the manuscript presented in an intelligible fashion and written in standard English?

Reviewer #1: Yes

Reviewer #2: Yes

6. Review Comments to the Author

Reviewer #1: Title:

Prevalence and factors associated with Multidrug Resistant Escherichia coli carriage on chicken farms in west Nile region in Uganda: A cross-sectional survey.

Abstract

OK

Introduction

OK

Methods

Line 123-124: This sentence “On each farm, a household member who met the study's selection criteria was purposively chosen for 124 the interview” is repeated in the following sub heading (line 140-141).

State clearly in the methods that the analysis was phenotypic rather than genotypic.

Results

Line 292-294: Fig 1: Thanks for making it clear. Now it looks like a table. It should be Table 4 instead.

Discussions

OK

Conclusion

OK

General comments:

This is a very good study that add value to the literature. I am okay for it to be published after fixing these very minor comments.

Reviewer #2: The article meets the criteria for PLOS Global public health journal publication and has a technically sound manuscript with conclusions that are supported by data presented in the quantitative tables. The cross sectional design for the primary research question posed projects an ethically and methodologically sound design with mentioned limitations and weaknesses.

This is further elaborated by the drawn conclusions which even with the mentioned recall bias in the design, not only answers the question posed but go on to identify factors associated with and potentially propagating multi drug resistance among chicken farms in Uganda. Despite the mentioned practical hindrances to implementing antibiotic resistance awareness among female Uganda chicken farmers, the results and conclusion clearly show and advocates for such a vital public health intervention particularly in the developing nation. The conclusions also supports this notion.

An astounding statistical analysis of the data is seen through the sub-categorization of data captured from the snapshot survey and responses from the questionnaire. The mathematical formulas used are clearly shown and applied. Furthermore, the manuscript is presented in good palatable english.

7. PLOS authors have the option to publish the peer review history of their article (what does this mean?). If published, this will include your full peer review and any attached files.

**Do you want your identity to be public for this peer review?** For information about this choice, including consent withdrawal, please see our Privacy Policy.

Reviewer #1: No

Reviewer #2: **Yes: **Benson Tarisai Gombe

---

## [Editor Report · Decision Letter 2]

20 Dec 2024

Prevalence and factors associated with multidrug resistant Escherichia coli carriage on chicken farms in west Nile region in Uganda: A cross-sectional survey

PGPH-D-24-01295R2

Dear Doctor Nyolimati,

We are pleased to inform you that your manuscript 'Prevalence and factors associated with multidrug resistant Escherichia coli carriage on chicken farms in west Nile region in Uganda: A cross-sectional survey' has been provisionally accepted for publication in PLOS Global Public Health.

Best regards,

Ben Pascoe

Academic Editor

Thank you for incorporating feedback from the reviewers into a revised version of the manuscript, your work can now be accepted for publication.